# War-related vicarious trauma among healthcare providers in the war-torn tigray, Northern Ethiopia

Hagos Degefa Hidru[1]*, Mengistu Hagazi Tequare[1,2], Abadi Kidanemariam Berhe[1,3], Gidey Gebremeskel Kidane[4], Yemane Gebremariam Gebre[5], Mohamedawel Mohamedniguss Ebrahim[4], Aregawi Gebreyesus[2], Gebregziabher Berihu Gebrekidan[2], Reiye Esayas Mengesha[4], Gebrekiros Gebremichael Meles[2], Assefa Ayalew Gebreslassie[2], Bereket Berhe Abreha[4], Desalegn Massa Teklemichael[2], Berihu Gidey Aregawi[3], Haftom Tesfay Gebremedhin[6], Gebremedhin Gebreegziabher Gebretsadik[3], Yemane Berhane Tesfau[3], Gebreyesus Elfu Mezgebe[7], Zemichael Weldegebriel Asreshey[8], Girmay Medhin[9], Hailay Gebretnsae[1], Yibrah Alemayehu Haile[10], Tedros Gobezay Desta[10], Tsegay Berihu Tesfay[10], Alem Gebremariam[1,3]

1 Tigray Health Research Institute, Mekelle, Tigray, Ethiopia, 2 School of Public Health, College of Medicine and Health Sciences, Mekelle University, Mekelle, Tigray, Ethiopia, 3 Department of Public Health, College of Medicine and Health Sciences, Adigrat University, Adigrat, Tigray, Ethiopia, 4 School of Medicine, College of Health Sciences, Mekelle University, Mekelle, Tigray, Ethiopia, 5 School of Medicine, College of Medicine and Health Sciences, Adigrat University, Adigrat, Tigray, Ethiopia, 6 Department of Psychiatry, College of Medicine and Health Sciences, Adigrat University, Adigrat, Tigray, Ethiopia, 7 School of Medicine, College of Medicine and Health Sciences, Aksum University, Aksum, Tigray, Ethiopia, 8 School of Public Health, College of Medicine and Health Sciences, Aksum University, Aksum, Tigray, Ethiopia, 9 Aklilu Lemma Institute of Pathobiology, Addis Ababa University, Addis Ababa, Ethiopia, 10 Tigray Health Bureau, Mekelle, Tigray, Ethiopia

* hagosdeg@gmail.com

## Abstract

### Background

Tigray War, which started in November 2020, has doubled the burden on health-care professionals, causing both direct violence and indirect effects as they care for trauma survivors. Vicarious trauma refers to harmful changes that occur in professionals' views of themselves and/or others as a result of deep empathic engagement and repeated exposure to details of trauma survivors. There is insufficient data to determine the prevalence of vicarious trauma among healthcare workers in the region. Therefore, this study assessed the extent of vicarious trauma among health-care providers working in the war-torn Tigray region of Ethiopia.

### Method

A health facility-based cross sectional study design was used to recruit 2,374 health-care providers from August to September 2023 in the war affected Tigray region of Ethiopia and the study participants were selected using stratified random sampling

**Data availability statement:** All relevant data are within the paper and its Supporting Information files.

**Funding:** The author(s) received no specific funding for this work.

**Competing interests:** No authors have competing interests.

techniques. Data were collected using the Open Data Kit (ODK) and exported to SPSS version 23.0 for analysis. Vicarious trauma was assessed using a 7-item standard tool with likert scale responses ranging from very often (5) to rarely (0). The total score from the responses on these 7 items was categorized as low risk (0–14), moderate risk (15–21), high risk (22–28), and extremely high risk (29–35). Ordinal multivariable logistic regression was used to identify the association of various factors with vicarious trauma. Statistical significance was reported when p-value was less than 0.05.

## Result

Prevalence of moderate and above vicarious trauma was revealed as 81.6% (95% CI: 80.0, 83.1). Specifically, the burden was identified as 32.4%, 33.3%, and 15.8% for moderate, high,and extremely high levels of vicarious trauma respectively, with significant variability across the administrative zones of the region. Risk of vicarious trauma increases with increasing age 30–39 years (AOR = 1.3, p-value = 0.002), 40–49 years (AOR = 1.5, p-value = 0.01), 50 years and above (AOR = 2.3, p-value = 0.0001), having a larger family size (AOR = 1.2, p-value = 0.03), among those who live far from their work place (AOR = 1.3, p-value = 0.04), and among those in leadership positions (AOR = 1.5, p-value = 0.0001).

## Conclusion

Healthcare workers in Tigray region experienced high level of vicarious trauma (8 out of 10). This was worse for older workers with larger families, those living and walking farther to work, and those working in heavily conflict-affected central and northwest Tigray. This calls for the government and stakeholders urgently collaborate to provide training on mental resilience, coping strategies, and support resources, offer age-specific psychoeducation, consider adjusting work hours for older employees, support affected employees and their families, improve transportation, and reduce administrative burdens on managers to prioritize staff well-being.

---

## Introduction

Vicarious traumatization (VT) refers to harmful changes that occur in professionals' views of themselves and others, as a result of exposure to the graphic and/or traumatized psychological needs, beliefs, and memory systems of their clients who treat trauma survivors [1,2]. The war that broke out in Tigray, Northern Ethiopia, in November 2020 has looted and vandalized more than $3.78 billion in total cost in monetary terms and the destruction of economic value exceeded $2.31 billion. It has taken a toll on the physical, mental, economic, and psychological health of the people in Tigray in general and the healthcare providers in particular [3,4].

Many healthcare workers have been killed, displaced, or have left their posts due to fear, lack of payment, and the extremely stressful and dangerous working conditions [5]. Witnessing horrific injuries, dealing with mass casualties, and the constant

fear for their safety have led to depression, burnout, and vicarious trauma among healthcare providers [6]. There is a significant need for mental health and psychosocial support for both the affected population and healthcare workers who have experienced trauma [7]. Exposure to the profound suffering and traumatic narratives of those they serve places healthcare professionals and humanitarian aid workers in conflict zones at significant risk of developing vicarious trauma [8].

Healthcare professionals face significant risks during wartime due to a combination of direct and indirect factors. The deliberate targeting of healthcare systems leads to widespread disruption of services, while the overwhelming influx of war-wounded and trauma survivors places immense pressure on providers [9]. This constant exposure to the horrific realities of conflict, including the direct impact on their own lives and the suffering of their patients, makes them particularly vulnerable to vicarious trauma [3]. This form of trauma results either from deep involvemnet and or repeated exposure to the traumatic experiences of others, highlighting the profound psychological toll that war takes on those dedicated to healing [10]. The frequency of vicarious trauma, which is based on the knowledge that more exposure to traumatized clients and larger workloads raise the risk of this phenomena among health workers.

The prevalence of vicarious trauma varies, with a startling 90.3% among nurses in Japanese hospitals [11] and 12% among palliative care physicians in the USA [12]. Additionally, research conducted among child welfare professionals in the United States [13] revealed that between 26 and 35 percent of healthcare professionals experienced vicarious trauma.

A more comprehensive view from a systematic analysis in Australia showed that 56% [14] of healthcare personnel working in the alcohol and drug field had experienced secondary trauma. Similarly, research from the University of Ottawa and Poland showed that 50% of therapists and 43.4% of medical professionals had experienced secondary trauma, respectively [15,16]. Vicarious trauma was reported by a noteworthy 45% of medical staff during the Gaza conflict [17]. In Kenya, 67% of medical staff working on hospitals reported experiencing vicarious trauma [18], similarly a finding from systematic review indicated that vicarious trauma prevalences ranging widely from 26% to 90% [19] among health professionals working on caregivers.

As a result of the war in Ethiopia, hundreds of thousands of people have been domestically and internationally displaced, which has also had a devastating impact on supply lines and agricultural production [20]. There were severe food shortages because of blockades, especially on Tigray; estimates suggest that hundreds of thousands of people were in danger of starvation. In Tigray and other impacted areas, the health system was completely destroyed. Numerous hospitals, health centers, and health posts suffered significant damage, were looted, or were rendered inoperable [3]. Essential services like prenatal care, child vaccines, and emergency medical treatment were severely reduced as a result of the shortage of medical supplies and the targeting of healthcare personnel. Children missed school because of the attacks, occupations, and looting that occurred in schools. These diverse effects present a somber image of the war's aftereffects [3–6].

Despite our personal observations and experiences with such trauma among healthcare providers in the war-torn Tigray region, the scope of the VT is not adequately investigated except for a single recently published study on the presence of vicarious trauma among the medical staff in the hemodialysis unit of the Ayder referral hospital [9]. Thus, there are many other consequences of the war that need thoughtful consideration. Among these, the burden on health care providers and the extremely serious lack of access to medical supplies to treat survivors that often result from the war cannot be over-emphasized. Therefore, this facility-based cross-sectional study was conducted to estimate the extent of vicarious trauma among healthcare professionals in the Tigray region of Ethiopia.

## Methods and materials

### Study area and period

The study was conducted on public health facilities (hospitals, health centers, and health post) in the war-torn Tigray region of Ethiopia. Before the breakout of the war, the region had a well-structured three-tiered health system; a primary healthcare unit that includes health posts (used for basic healthcare service provided at the community level), health centres and

primary hospitals; secondary care provided by general hospitals; and tertiary care provided by specialized referral hospitals [21]. As of November 2020, the region had 40 hospitals (2 specialized referral, 14 general, 24 primary), 226 health centers, and 741 health posts [22,20]. Healthcare provider is a licensed individual, organization, or institution that delivers healthcare services. Before the war, there were 19,324 health workers in total, excluding the two federally run referral hospitals, Ayder (Mekelle) and Axum. These included specialist physicians (69), general practitioners (411), nurses (4,422), midwives (1,394), pharmacists (296), laboratory specialists (494), health officers (935), public health specialists (431), HEWs (1,918), supportive staff (5,344, and others (4,018) [15]. Immediately after the war, 514 (80.6%) health posts, 153 (73.6%) health centers, 16 (80%) primary hospitals, 10 (83.3%) general hospitals, and 2 (100%) specialized hospitals were damaged and/or vandalized either fully or partially due to the war [23]. The study was conducted from August to September 2023.

## Study design and population

A facility-based cross-sectional study design was conducted and all randomly selected health care providers working in Tigray region public health facilities during the war was included in this study.

## Sample size and sampling procedure

The sample size was estimated considering a confidence level of 95% (Z = 1.96), a margin of error of 0.03, a 50% prevalence of vicarious trauma among healthcare workers, a 10% non-response rate and a design effect of 2. This resulted in a final sample size of 2374. A stratified multistage sampling technique was used to reach each health facility. We have used p 50% because there were no related studies in Ethiopia and taking prevalence from other countries might not represent the situation in Tigray. Thus,we used p 50%. Regarding the design effect of 2, as a rule of thumb design effect of 2 or 1.5 is frequently used in sampling, especially for cluster, multistage, and stratified designs because of their stages and intracluster correlation, to provide a practical way to inflate the sample size and account for the potential loss of precision compared to simple random sampling. However, it is essential to recognize its limitations and strive for more precise estimates of the design effect whenever possible. Finally a total of 2374 study participants were proportionally allocated to each of the selected health facilities and then a simple random sampling technique was used to recruit individual potential respondents within the study health facilities (Fig 1).

## Data collection tool and method

A total of seven items from the Crisis and Trauma Resource Institute's (CTRI) tool developed by (Vicki Enns, vol02, 2020) were used to evaluate vicarious trauma among healthcare providers. The vicarious trauma-related 7-item questionnaire with responses on a Likert scale ranging from very often (5) to rarely (0) was used to measure vicarious trauma. To estimate the degree of vicarious trauma, the scores were added up; larger scores indicate a greater degree of vicarious trauma. The CTRI classifies trauma with a cumulative score as low risk (0–14), moderate risk (15–21), high risk (22–28), and extremely high risk (29–35) [24]. In addition, the tool includes socio-demographic and occupation related characterstics.

The data was collected through a face-to-face interviewer-administered questionnaire. After extensive revision of the socio demographic and work environment related characterstics of the English questionnaire, the final English version was translated into the local language by language experts. Supervisors and data collectors were trained on the objective of the study, instructions for the method, how to obtain informed consent, how to approach participants, ethical procedures, and general information on vicarious trauma. Data were collected using twelve Master holder health professionals fluent in the local language (Tigrigna).

## Data quality control

The questionnaire was pre-tested on 5% of the study population in the non-selected health facilities to check the open data kit (ODK) completeness and to ensure clarity, wording, logical sequence, and skip patterns of the

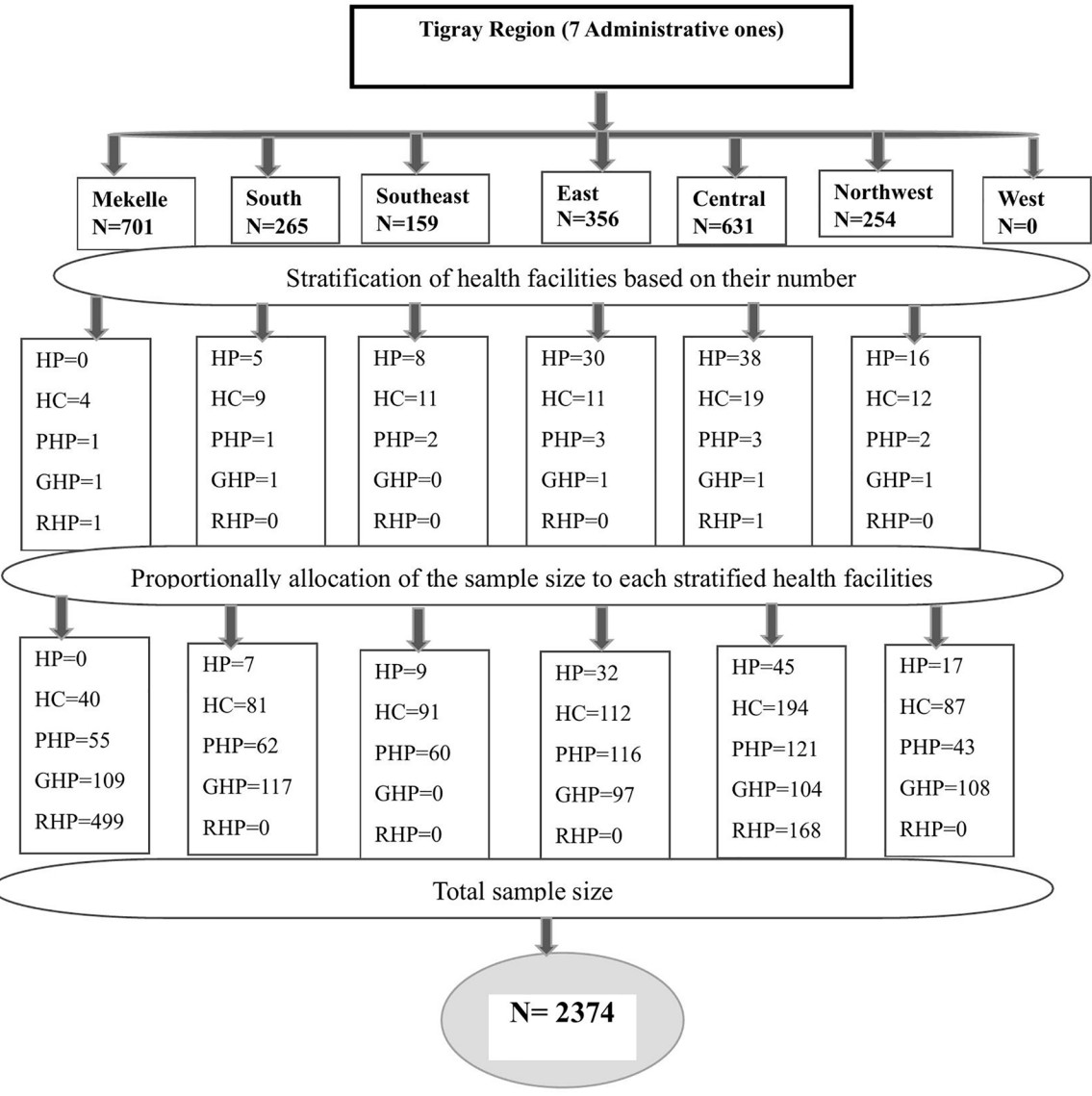

Figure 1: Schematic presentation of sampling technique and procedures to select the health facilities and health professionals in the war torn Tigray, Ethiopia, 2023.

**Fig 1. Schematic presentation of sampling technique and procedures to select the health facilities and health professionals in the war torn Tigray, Ethiopia, 2023.**

questions. Appropriate revison were made after discussing them with the supervisor before starting the actual data collection process. Data collection location, accuracy, and completeness were monitored by the data manager throughout the data work flow using the ODK data repository. In addition, every day the filled-out questionnaire is checked before a respondent leaves the setting by data collectors and supervisors. The reliability and validity of the outcome tool measured by a Cronbach's alpha and McDonald's Omega (ω), respectively was found 0.865 and 0.868.

## Data processing and analysis

Data were exported from ODK to SPSS version 23.0 for cleaning and statistical analysis. Descriptive statistical methods such as frequencies and percentages were used to summarize categorical data, while the mean and the standard deviation were used for continuous variables. The association of the independent variables with the level of vicarious trauma (low risk, moderate risk, high risk, and extreme high risk) was investigated using ordinal logistic regression. Our study, leveraging a substantial sample size that inherently yielded a high variable-to-case ratio, all variables having ($p < 0.05$) on the bivariate ordinal logistic regression analysis were candidates for the ordinal multivariable logistic regression analysis to control confounders. This approach was deliberately employed to rigorously control for potential confounding influences and to examine the relationship between independent and dependent variables. Statistical significance was reported whenever p-value $\leq 0.05$, and the strength of the association was quantified using the adjusted odds ratio (OR) along with a 95% confidence interval. A multi-collinearity test was conducted to see any correlation between the independent variables using the co-linearity diagnostic test (VIF and tolerance test). The model assumptions for goodness of fit were checked and fitness was declared at p-value $\geq 0.05$. The estimated sample size and implied power analysis relate to measuring significance in the primary outcome of vicarious trauma only, not to the regression model. When we report our results of multivariable ordinal logistic regression, we used the term "higher level vicarious trauma," which means extreme high versus combined low, medium, and high; or combined "high and extreme high" versus combined low and medium; or combined "medium, high, and extreme high" versus low.

## Ethical consideration

Ethical approval and clearance were obtained from the Mekelle University Institutional Review Board (IRB) with a reference number (MU-IRB 2049/2023). Then the Tigray Regional Health Bureau (TRHB) secured a letter of cooperation from each of the zonal and district administrations of the selected health facilities. After being informed of the aim of the study by data collectors and supervisors, written consent was obtained from each study participant. Confidentiality of the information is assured, and privacy is respected and kept as well.

## Result

### Socio-demographic characteristics of study participants

Of 2374 participants, 2366 were participated in this study, resulting in a response rate of 99.7%, and 60.7% of these respondents were female. The respondents' mean age was 34 years (SD=±8) and more than 80% were within the age range of 20–39 years. Two-thirds of the study participants were married and had more than three family members. The majority (96.4%) of the study participants were followers of the Christian orthodox religion. Above half (56.3%) of the study participants were from the Mekelle and Central zones, and about three-fourths worked in hospital settings. The participants had a median (IQR) work experience of 8.0 (5.0, 12.0) years and a median annual income (IQR) of 8017 Ethiopian Birr (range: 7071.0–9056.0). Above two-third (66.1%) of the respondents were permanently living with their parents. Almost half (48.0%) of the respondents were nurses, and about 1452 (61.4%) of the respondents had a BSc degree. The majority, 94.3% of the study participants, had no alternative source of income, and more than half perceived themselves as economically poor. More than three-fourths, 79.0%, reported working more than or equal to 40 hours per week, and more than half, 54.1%, occasionally worked on weekends. About half (50.4%), of the healthcare providers took less than 25 minutes to reach their working health facilities (Table 1).

### Prevalence of vicarious trauma

The overall prevalence of vicarious trauma (moderate and above) was found to be 81.6% [95% CI: 80.0, 83.1], with their specific level indicated as moderate, high, and extremely high accounting for 32.4%, 33.3%, and 15.8%, respectively (Fig 2).

**Table 1. Socio-demographic and work-related characteristics of the study participants in Tigray, Ethiopia, from August to September 2023.**

| Variables (N = 2366) | Categories | Frequency | Percentage (%) |
|---|---|---|---|
| **Zone address** | Mekelle | 701 | 29.7 |
| | Central | 631 | 26.6 |
| | Eastern | 355 | 15.1 |
| | North-west | 254 | 10.7 |
| | South-east | 265 | 11.2 |
| | South | 159 | 6.7 |
| **Age (in year)** | 20-29 | 864 | 36.5 |
| | 30-39 | 1052 | 44.5 |
| | 40-49 | 211 | 8.9 |
| | 50+ | 239 | 10.1 |
| **Sex of the respondents** | Male | 930 | 39.3 |
| | Female | 1436 | 60.7 |
| **Marital status** | Never married | 666 | 28.1 |
| | Currently married | 1538 | 65.0 |
| | Divorced/Widowed/Separated | 162 | 6.8 |
| **Family size** | 1-2 | 694 | 29.3 |
| | 3-4 | 999 | 42.2 |
| | ≥5 | 673 | 28.4 |
| **Permanent residence of family members** | Living together | 1565 | 66.1 |
| | Living separately | 801 | 33.9 |
| **Have your own house** | Yes | 678 | 28.7 |
| | No | 1688 | 71.3 |
| **Type of living house** | Rent house | 1661 | 70.2 |
| | Own house | 450 | 19.0 |
| | Family house | 165 | 7.0 |
| | Governmental house | 90 | 3.8 |
| **Perceived economic classification** | Rich | 8 | 0.3 |
| | Medium | 912 | 38.5 |
| | Poor | 1342 | 56.7 |
| | Unable to classify | 79 | 3.3 |
| | I do not want to talk | 25 | 1.1 |
| **Current profession** | Laboratory | 160 | 6.8 |
| | Pharmacy | 175 | 7.4 |
| | Medical doctor | 182 | 7.7 |
| | Nurse | 1135 | 48.0 |
| | Midwife | 297 | 12.6 |
| | Public health officer | 120 | 5.1 |
| | Health Extension worker | 145 | 6.1 |
| | Others | 152 | 6.4 |
| **Educational level** | Level 3/4 or Diploma | 645 | 27.3 |
| | BSc Degree | 1452 | 61.4 |
| | MSc/MPH | 87 | 3.7 |
| | Medical Doctor | 98 | 4.1 |
| | Specialist/Sub-specialist | 84 | 3.6 |
| **Managerial position** | Yes | 441 | 18.6 |
| | No | 1925 | 81.4 |

*(Continued)*

**Table 1.** (Continued)

| Variables (N = 2366) | Categories | Frequency | Percentage (%) |
|---|---|---|---|
| **Total work experience** | <5 years | 1036 | 43.8 |
| | 5 to 9 years | 952 | 40.2 |
| | 10 years or above | 378 | 16.0 |
| **Alternative source of income** | Yes | 134 | 5.7 |
| | No | 2232 | 94.3 |
| **Work at weekends** | Always | 652 | 27.6 |
| | Sometimes | 1279 | 54.1 |
| | Rarely | 130 | 5.5 |
| | Not at all | 305 | 12.9 |
| **Work at night shift** | Always | 298 | 12.6 |
| | Sometimes | 1431 | 60.5 |
| | Rarely | 115 | 4.9 |
| | Not at all | 522 | 22.1 |
| **Means of travel to work** | Taxi/Bajaj/Public Transport | 236 | 10.0 |
| | Facility car | 351 | 14.8 |
| | On foot | 1737 | 73.4 |
| | Others | 42 | 1.8 |
| **Distance from home to workplace in minutes** | <=25 | 1192 | 50.4 |
| | 26-50 | 691 | 29.2 |
| | 51-75 | 239 | 10.1 |
| | >=76 | 244 | 10.3 |
| **Monthly gross salary in ETB** | =<7071 | 1036 | 43.8 |
| | 7072-9056 | 1018 | 43.0 |
| | >=9057 | 312 | 13.2 |
| **Working hours per week** | <40 hours | 1550 | 65.5 |
| | >=40 hours | 816 | 34.5 |

### Factors associated with vicarious trauma

Eight variables (zone, age category, family size, management position, working hours, and walking distance to the health facility) were included in the multivariable ordinal logistic regression model, and all variables, except working hours, were found to be significantly associated with vicarious trauma.

There was a higher level of vicarious trauma among healthcare providers working in the north-west (AOR = 1.5; 95% CI: 1.1, 2.0) and central (AOR = 1.4; 95% CI: 1.1, 1.7) zones compared to the Mekelle zone. Compared to healthcare providers in their age groups of 20–29 years, the odds of being at a higher level of vicarious trauma were higher as age increased from 30–39 years (AOR = 1.3, 95% CI: 1.1, 1.6), 40–49 years (AOR = 1.5, 95% CI: 1.1, 2.0), and 50 years and older (AOR = 2.3, 95% CI: 1.7, 3.0). The odds of being at a higher level of vicarious trauma were 1.5 times higher among those with management positions as compared to healthcare providers who were not in management positions (AOR = 1.5, 95% CI: 1.2, 18). The odds of experiencing a higher level of vicarious trauma were 1.2 times higher among healthcare providers with family sizes of five and above as compared to those with family sizes of less than five (AOR = 1.2, 95% CI: 1.1, 1.5).

The likelihood of experiencing a higher level of vicarious trauma was 1.3 times higher among healthcare providers walking a distance greater than 60 minutes from home to their health facility than healthcare providers whose walking distance is less than 30 minutes (AOR = 1.3, 95% CI: 1.1, 1.6) (Table 2).

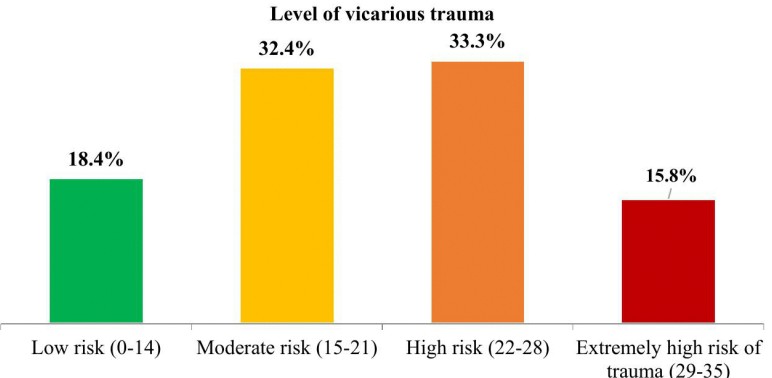

**Fig 2. Level of vicarious trauma among health care professionals in the war-torn Tigray, North Ethiopia, August to September, 2023.**

**Table 2. Factors associated with vicarious trauma among healthcare providers in Tigray, Ethiopia, August to September, 2023.**

| Independent Variables (N = 2366) | | Vicarious trauma, n (%) | | | | COR [95% CI] | AOR [95% CI] |
|---|---|---|---|---|---|---|---|
| | | Low | Moderate | High | Extreme high | | |
| Zone address | Mekelle | 129 (18.4) | 251 (35.8) | 225 (32.1) | 96 (13.7) | Ref. | Ref. |
| | Central | 99 (15.7) | 179 (28.4) | 215 (34.1) | 138 (21.9) | 1.5 [1.2, 1.8] | 1.4 [1.1, 1.7]* |
| | Eastern | 73 (20.5) | 104 (29.2) | 120 (33.7) | 59 (16.6) | 1.1 [0.9, 1.4] | 1.0 [0.8, 1.3] |
| | North west | 25 (9.8) | 65 (25.6) | 105 (41.3) | 59 (23.2) | 2.0 [1.6, 2.6] | 1.5 [1.1, 2.0]* |
| | South east | 40 (25.2) | 63 (39.6) | 49 (30.8) | 7 (4.4) | 0.6 [0.5, 0.9] | 0.7 [0.5, 0.9]* |
| | South | 70 (26.4) | 105 (39.6) | 74 (27.9) | 16 (6) | 0.6 [0.5, 0.8] | 0.8 [0.6, 1.0] |
| Age in years | 20-29 | 206 (23.8) | 299 (34.6) | 265 (30.7) | 94 (10.9) | Ref. | Ref. |
| | 30-39 | 174 (16.5) | 347 (33.0) | 355 (33.7) | 176 (16.7) | 1.5 [1.28, 1.77] | 1.3 [1.1, 1.6]* |
| | 40-49 | 37 (17.5) | 57 (27) | 77 (36.5) | 40 (19.0) | 1.7 [1.30, 2.27] | 1.5 [1.1, 2.0]* |
| | 50+ | 19 (7.9) | 64 (26.8) | 91 (38.1) | 65 (27.2) | 2.8 [2.19, 3.70] | 2.3 [1.7, 3.0]* |
| Family size | 1-4 | 342 (202) | 561 (33.1) | 559 (33.0) | 231 (13.6) | Ref. | Ref. |
| | ≥ 5 | 94 (14.0) | 206 (30.6) | 229 (34.0) | 144 (21.4) | 1.5 [1.29, 1.79] | 1.2 [1.1, 1.5]* |
| Management position | Yes | 75 (17.0) | 109 (24.7) | 162 (36.7) | 95 (21.5) | 1.5 [1.22, 1.79] | 1.5 [1.2, 1.8]* |
| | No | 361 (18.8) | 658 (34.2) | 626 (32.5) | 280 (14.5) | Ref. | Ref. |
| Working hour | <40 | 273 (17.6) | 482 (31.1) | 528 (34.1) | 267 (17.2) | Ref. | Ref. |
| | ≥ 40 | 163 (20.0) | 285 (34.9) | 260 (31.9) | 108 (13.2) | 0.8 [0.68, 0.92] | 1.0 [0.8, 1.1] |
| Walking distance | <30 | 252 (21.1) | 387 (32.5) | 387 (32.5) | 166 (13.9) | Ref. | Ref. |
| | 30-59 Minute | 111 (15.9) | 229 (32.9) | 250 (35.9) | 107 (15.4) | 1.2 [1.05, 1.46] | 1.1 [0.9, 1.3] |
| | 60 + minutes | 73 (15.3) | 151 (31.7) | 151 (31.7) | 102 (21.4) | 1.4 [1.19, 1.75] | 1.3 [1.1, 1.6]* |

AOR = Adjusted Odds Ratio, COR = Crude Odds Ratio, CI = Confidence Interval, * = p < 0.05.

## Discussion

This study assessed the prevalence of vicarious trauma among healthcare providers in the war-torn Tigray region using the Crisis and Trauma Resource Institute's (CTRI) classification with a cumulative score of low risk (0–14), moderate risk (15–21), high risk (22–28), and extremely high risk (29–35). More than eighty percent of the healthcare providers had vicarious trauma, and it was significantly higher among older age, with a larger family size, walking sixty minutes and above to their work place, with managerial positions, and variation across working zones.

The prevalence of vicarious trauma (81.6%) in our study revealed a high number of healthcare providers suffering from moderate and severe vicarious trauma. This finding was higher than studies done in Kenya (67%) (18), Arizona (61.3%) [25], Gaza (45%) (17), Poland (43.4%) (16), the United States of America (12%) (12), (35%) (13), and Canada (50%) [26]. This disparity may have resulted from the war in Tigray, which raised the risk of violence and was characterized by great cruelty, reports of killings, heinous sexual violence, and immense health needs created by the conflict. It could also be the result of frequent exposure to or deep interaction with the severely impacted survivors.

In addition, different evidences from the humanitarian agenecies is supported to this finding; as Medicine Sans Frontier (MSF) has repeatedly said, medical staff and facilities must be protected throughout a conflict in accordance with international humanitarian law; this is clearly not happening in Tigray [27]. Professionals in the fields of education and healthcare who live and work in conflict environments sustain psychological harm that can significantly affect their lives and careers [6]. The emotional strain on healthcare workers, including feelings of sadness, exhaustion, and trauma from witnessing others' suffering, is pointed out by Physicians for Human Rights (PHR) and others who have reported serious human rights abuses, like attacks on hospitals and sexual violence during conflicts that continued even after the Cessation of Hostilities Agreement (CoHA) was signed in November 2022 (5). Instead of helping those affected by the conflict, medical staff have told reporters that they are now just issuing death certificates. Their work is made harder by a lack of equipment and supplies. Healthcare providers sadly mentioned that they "are sending patients home to die," because they cannot provide treatment in these conditions. Many healthcare workers are starting to go hungry because they haven't gotten their government salaries in a while [28].

Consequently, medical personnel were forced to treat a high volume of patients with the potentially dangerous physical and psychological effects of the conflict on a regular basis. Another explanation could be that underfunding of, damage to, and theft from healthcare institutions have led to critical shortages in medical experts and essential medical supplies needed for treatment [29,30]. This places a great deal of strain on the remaining employees, who are already probably overburdened and worn out, making them even more susceptible to vicarious stress. The Tigray war was characterized by ethnic violence and the targeting of civilians. Medical professionals who have experienced personal loss, have lost loved ones or families, or have been uprooted themselves may be re-traumatized when tending to patients who have also experienced such trauma. This may exacerbate their inability to detach themselves from their traumatized patients and heighten their sense of helplessness and hopelessness.

The odds of experiencing a higher level of vicarious trauma among healthcare providers were higher as they went from Mekelle to the Central and Northwest zones. This might be because healthcare providers working in these regions were exposed to more violence and trauma, heavy fighting during the war, shortages of medical supplies as a result of the war, may have lost colleagues, friends, or family members during the war, and/or may have met and provided care for people who had been injured by atrocities such as the mass massacre in Tsion, Aksum on November 29, 2020 by troops of Ertirea and Amhara milishas [4]. As a result, the looted health system and more traumatized people demanding healthcare providers were high, making them worried.

Similarly, the odds of experiencing a higher level of vicarious trauma were high as the age group increased from 20–29–30–39 years, 40–49 years, and 50 + years, which is in line with the study done in Kenya [18]. This might be because of increasing life stage responsibilities and associated personal stressors among care providers over 30 such as family obligations, responsibilities to communities, financial obligations, or aging parents, making them less resilient to additional emotional burdens from work leading to increased risk of vicarious trauma when caring for traumatized clients or patients. It might also because of the age differences in stress and coping processes, this means that over time, older adults used less confrontive coping and more positive reappraisal and avoidance coping strategies [31].

In addition, the odds of being at a higher level of vicarious trauma were high among healthcare providers with a family size of five or above as compared to those with a family size less than five. This might be, in a war zone, basic resources (food, water, safety) are scarce. A healthcare provider with a large family must juggle the high-stakes survival needs of

their dependents with the traumatic demands of their clinical work. It also might be, larger families in conflict zones often mean more members exposed to trauma. The provider not only processes their own and their patients' trauma but also the secondary trauma of their family members. As a result, limited personal time and reduced opportunities for self-care and emotional processing due to family commitments might exacerbate the impact of vicarious trauma [32]. The odds of having a higher level of vicarious trauma were high among healthcare providers in management positions as compared to those without postion. This might be due to, managers/facility leaders were witnessed the overall impact of the war (displaced staff, lack of supplies), which creates a cumulative effect of trauma that clinical staff who focus on patients at a time might not see in its entirety. As a result, they might exposed to the collective trauma of their entire staff. Additionally, it might be difficult for managers to make decisions about resource allocation, staffing, and patient care under perilous circumstances, especially if they alone bear the burden of knowing traumatizing information about the healthcare system such as displaced staff, perilous medical supply, traumatized patients, and witnessing the overall impact of the war on healthcare delivery [6].

Furthermore, a higher level of vicarious trauma was experienced among healthcare providers who had to walking a distance of sixty minutes or more to their health facility. This might be walking long distances (over 60 minutes) and increased vicarious trauma is not merely a matter of physical fatigue, but a complex intersection of environmental exposure and professional helplessness [33]. Walking through war-torn areas acts as a continuous exposure to the sights and sounds of conflict, which serves as a physiological and emotional stressor even before the healthcare provider reaches the clinic. As a result, this constant state of hyper-vigilance during transit lowers the provider's psychological resilience, making them more susceptible to vicarious trauma when they eventually encounter patient suffering. The lack of fuel and transportation (necessitating the long walk) is often accompanied by the absence of banking and supplies. Thus, the provider arrives exhausted to find critically ill patients they cannot treat due to these blockades, they experience a profound sense of moral injury,the psychological distress resulting from actions, or lack thereof, that violate one's moral code (being unable to save a life due to systemic failure). Additionally, the ban on communication and absence of salaries during the war period removes the provider's own support system and safety net, leaving them with no emotional outlet for the trauma they witness [3].

## Strength and limitation

This study covered public health facilities in all zones of the war-torn Tigray region except the western and part of the southern zones because of security issues and had a large sample size. In addition, we tried to reveal the timely evidence the magnitude of vicarious trauma during the aftermath consequence of the Tigary war among healthcare professionals. As limitation in this study, private facilities were not included, did not consider the duration of the exposure with the war survivors and it is not tried to related for each type of the causes with vicarious trauma. Recall and social desirable biases might be introduced, especially for those affected by an extremely high risk of vicarious trauma. Since the variables are chosen for a model based on observed relationships in the data (i.e., empirically derived) rather than being driven by a clear, pre-existing hypothesis, researchers are more likely to stumble upon spurious correlations simply by chance and might be increased type I error.

## Conclusion and recomendation

Eight out of ten healthcare workers in Tigray were found to have experienced vicarious trauma as a result of the war. This rate was found to be significantly higher in those who were older, had larger families, lived farther away from their place of employment, and traveled on foot, and worked in the areas that saw frequent and intense fighting in the central and northwest of the region, which is far from the capital city than the Mekelle Zone. The health workforce's high rate of vicarious trauma may be detrimental to the standard of care provided to the population. Thus, the government ought to act right away to address the underlying causes of the suffering that medical professionals witness, working in tandem with

other stakeholders. In order to address the high rate of vicarious trauma, we advise the government and stakeholders to; provide training and capacity building for all employees regarding mental resilience, proactive coping strategies, vicarious trauma, and available support resources, age-specific psychoeducation and coping strategies; investigate options for adjusting work hours for older employees, especially those exhibiting signs of vicarious trauma, provide information and support to the affected employee and their family, along with guidance on family self-care, implement better transportation options for employees traveling long distances, especially those on foot, review and, if feasible, lessen the administrative burden for managers, freeing them up to concentrate on staff well-being and their own self-care.

## Supporting infomation

**S1 File. Vicarious trauma N = 2366 Labelled.**
(XLSX)

## Author contributions

**Conceptualization:** Hagos Degefa Hidru, Abadi Kidanemariam Berhe, Yemane Gebremariam Gebre, Reiye Esayas Mengesha, Assefa Ayalew Gebreslassie, Bereket Berhe Abreha, Gebremedhin Gebreegziabher Gebretsadik, Yemane Berhane Tesfau, Gebreyesus Elfu Mezgebe, Zemichael Weldegebriel Asreshey, Girmay Medhin, Alem Gebremariam, Tedros Gobezay Desta, Tsegay Berihu Tesfay.

**Data curation:** Hagos Degefa Hidru, Mengistu Hagazi Tequare, Gidey Gebremeskel Kidane, Mohamedawel Mohamedniguss Ebrahim, Reiye Esayas Mengesha, Desalegn Massa Teklemichael, Haftom Tesfay Gebremedhin, Gebremedhin Gebreegziabher Gebretsadik, Yemane Berhane Tesfau, Gebreyesus Elfu Mezgebe, Hailay Gebretnsae, Alem Gebremariam.

**Formal analysis:** Hagos Degefa Hidru, Abadi Kidanemariam Berhe, Mohamedawel Mohamedniguss Ebrahim, Gebregziabher Berihu Gebrekidan, Gebrekiros Gebremichael Meles, Alem Gebremariam.

**Investigation:** Assefa Ayalew Gebreslassie, Desalegn Massa Teklemichael, Berihu Gidey Aregawi, Zemichael Weldegebriel Asreshey.

**Methodology:** Hagos Degefa Hidru, Mengistu Hagazi Tequare, Mohamedawel Mohamedniguss Ebrahim, Aregawi Gebreyesus, Gebrekiros Gebremichael Meles, Assefa Ayalew Gebreslassie, Desalegn Massa Teklemichael, Haftom Tesfay Gebremedhin, Zemichael Weldegebriel Asreshey, Girmay Medhin, Hailay Gebretnsae, Alem Gebremariam.

**Project administration:** Yibrah Alemayehu Haile, Tedros Gobezay Desta, Tsegay Berihu Tesfay.

**Resources:** Yemane Gebremariam Gebre, Bereket Berhe Abreha, Gebreyesus Elfu Mezgebe, Yibrah Alemayehu Haile, Tedros Gobezay Desta, Tsegay Berihu Tesfay.

**Software:** Hagos Degefa Hidru, Mohamedawel Mohamedniguss Ebrahim, Gebrekiros Gebremichael Meles, Girmay Medhin.

**Supervision:** Hagos Degefa Hidru, Yemane Gebremariam Gebre, Yemane Berhane Tesfau, Girmay Medhin.

**Validation:** Mengistu Hagazi Tequare, Abadi Kidanemariam Berhe, Gidey Gebremeskel Kidane, Mohamedawel Mohamedniguss Ebrahim, Gebregziabher Berihu Gebrekidan, Gebrekiros Gebremichael Meles, Berihu Gidey Aregawi, Haftom Tesfay Gebremedhin, Gebreyesus Elfu Mezgebe, Girmay Medhin, Alem Gebremariam, Tsegay Berihu Tesfay.

**Visualization:** Mengistu Hagazi Tequare, Aregawi Gebreyesus, Alem Gebremariam.

**Writing – original draft:** Hagos Degefa Hidru, Mengistu Hagazi Tequare, Alem Gebremariam.

**Writing – review & editing:** Hagos Degefa Hidru, Abadi Kidanemariam Berhe, Gidey Gebremeskel Kidane, Yemane Gebremariam Gebre, Mohamedawel Mohamedniguss Ebrahim, Aregawi Gebreyesus, Gebregziabher

Berihu Gebrekidan, Assefa Ayalew Gebreslassie, Bereket Berhe Abreha, Berihu Gidey Aregawi, Gebremedhin Gebreegziabher Gebretsadik, Zemichael Weldegebriel Asreshey, Girmay Medhin, Hailay Gebretnsae, Alem Gebremariam.

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
