## [Decision Letter · Decision Letter 0]

20 Mar 2025

Dear Dr. Hidru,

Thank you for submitting your manuscript to PLOS ONE. After careful consideration, we feel that it has merit but does not fully meet PLOS ONE’s publication criteria as it currently stands. Therefore, we invite you to submit a revised version of the manuscript that addresses the points raised during the review process.

We look forward to receiving your revised manuscript.

Kind regards,

Selamawit Alemayehu Tessema

Guest Editor

PLOS ONE

Journal Requirements:

4. Your abstract cannot contain citations. Please only include citations in the body text of the manuscript, and ensure that they remain in ascending numerical order on first mention.

6. We note you have included a table to which you do not refer in the text of your manuscript. Please ensure that you refer to Table 2 in your text; if accepted, production will need this reference to link the reader to the Table.

Additional Editor Comments (if provided):

The authors raised an important topic and it addresses War related-Vicarious Trauma among Healthcare providers in the War-Torn Tigray,Northern Ethiopia. see specific comments below;

Abstract

• Avoid third-person writing “the”

• VT seems difficult to understand so it is better to define it in the abstract as well as in one sentence.

• “ we estimated “ third person writing the whole sentence needs rephrasing

• The burden of war and its impact especially on those with health providers, what is the need to do especially health care workers

• Interventions such as aftercare and addressing vulnerable groups

• I would put AOR of the factors significantly associated

• Sensitive topic more clarification on ethical consideration

Introduction

• Line 3-5 can you quantify the material or human damage with reference

• Citation for the “deliberate targeting of war on health care system causing disruption

• Paragraph 3; why healthcare professionals; are at risk needs rephrasing and synthesis

• Rather than listing all studies to connect ideas and flow in paragraphs 3 and 4

• First discussing issues in Western then Africa rather than shifting back and forth

Methodology

• You need to describe the desire why you used p 50 % inside the text and te design effect 2 ; need justification.

• If you put a diagram and the specific numbers when doing multi-stage sampling and proportional allocation

• What other tools or factors are assed other than VT which is not put in the methodology

• Why do you use a p-value 0.05 for bivariate logistic regression and put the justification as well

Result

• You can also put the reliability of the tools of outcome and validation in Ethiopia.

• The footnotes a, b,c are confusing as you only used p-value 0.05 as statically significant in your methodology.

• Why you did not asses the type of traumatic content disclosed to health providers

• “Consequently, medical personnel were forced to treat a high volume of patients with the potentially dangerous physical and psychological effects of the conflict on a regular basis. Another explanation could be that many medical experts and dangerous medical supplies needed for treatment and therapy have been forced to leave underfunded, damaged, and robbed healthcare institutions. This places a great deal of strain on the remaining employees, who are already probably overburdened and worn out, making them even more susceptible to vicarious stress. The Tigray war, which was characterized by ethnic violence and the targeting of civilians, which medical professionals may see personally while tending to patients, this could be because the victims have experienced personal loss, have lost loved ones or families, or have been uprooted themselves. This may exacerbate their inability to detach themselves from their traumatized patients and heighten their sense of helplessness and hopelessness.” This concept needs to be back up by citation and references

• The discussion needs to be improved as it does not compare or give appropriate citations for the explanation of the discussion points.

• I would add a specific and better recommendation.

Reviewers' comments:

Reviewer's Responses to Questions

**Comments to the Author**

1. Is the manuscript technically sound, and do the data support the conclusions?

Reviewer #1: Yes

Reviewer #2: Yes

Reviewer #3: Yes

Reviewer #4: Partly

Reviewer #5: Yes

2. Has the statistical analysis been performed appropriately and rigorously?

Reviewer #1: Yes

Reviewer #2: Yes

Reviewer #3: Yes

Reviewer #4: No

Reviewer #5: Yes

3. Have the authors made all data underlying the findings in their manuscript fully available?

Reviewer #1: No

Reviewer #2: Yes

Reviewer #3: Yes

Reviewer #4: No

Reviewer #5: Yes

4. Is the manuscript presented in an intelligible fashion and written in standard English?

Reviewer #1: No

Reviewer #2: No

Reviewer #3: Yes

Reviewer #4: No

Reviewer #5: Yes

Reviewer #1: Date: November 12, 2020

Dear Editor,

I hereby submit a review report of the manuscript, entitled ‘War related -Vicarious Trauma among Healthcare providers in the War-Torn Tigray, Northern Ethiopia.”. I acknowledge the efforts of the authors for performing a good job and writing evidence which is highly needed. Since, it needs further amendment, I suggested to be accepted with Major revision. I believe, this manuscript will be very helpful in passing better evidence for decision making in rebuilding the human resource development.

For the sake of simplicity, it would be good to create line numbers of the manuscript which would be easier for the authors to trace the comments and revised changes.

Abstract:

On the method section of the abstract a statement is described as “VT was assessed using a 7-item standard tool with likert scale…”, a phrase in a new line statement shall be written with full words of the phrase, otherwise, authors are advised to write Vicarious Trauma instead of VT.

On the second line of the result section of the abstract, authors need to rephrase the “was found to be”.

The conclusion of the abstract seems overlooked.

Introduction

On the first and second paragraphs: Authors have verified about the war in Tigray and its effects on the vulnerability of the health workers referring three peer reviewed articles. Ample literature written by humanitarian organizations have proven the health workforce traumatic experiences, so authors are requested to justify their statements from the perspective of humanitarian agencies.

Third and fourth paragraph, the piece of evidences from Gaza, Poland, Eastern Maynemar, Kenya, USA, Japan and findings from University of Ottawa needs to be consolidated, refined and synthesized. At this level, it would be good if the information could convey precise and concise message, otherwise it would be redundant. Authors are requested to revise the paragraph in a suitable manner for readers.

Though large number of health workforce population and civilians suffer in SSA countries due to war and conflict, the experiences and prevalence of vicarious trauma in sub Saharan African countries seem to be overlooked. So, authors are requested to provide SSA countries experiences.

On top of these comments, authors are advised to revise the flow and coherence of the paragraphs in the introduction section. It would be advisable to follow the writing style from global to local and general to specific.

Last paragraph of the introduction section: Authors noted “Despite our personal observations and experiences …….. the scope of the problem is not investigated”. The highlighted phrase seems overstated. Because, significant number of report reviews have been documented by MSF,

So, authors are advised to revise the phrase; instead of “not investigated” “the scope of the problem is not adequately investigated” might be an option. It would also be good to rephrase the valid evidence stated as “Except for the presence of vicarious trauma among the medical staff in the hemodialysis unit of the Ayder referral hospital published recently”.

Methods and materials

Study area

It is good that authors noted the number of health facilities and health workforce before the war. However, it seems inadequate information for this manuscript. On top of the information given the categories of the health workers, their speciality and the development of the health workforce and its contribution on the health outcome need to be highlighted.

Study design

The design is not only for the purpose of recruiting healthcare providers. It works for conducting the overall study. Authors are advised to omit the phrase “recruiting healthcare providers”.

Sample Size and Sampling Procedure

What was the reason for authors to choose simple random sampling technique after using multistage stratified sampling technique, while systematic random sampling technique is the easiest technique? It is from curiosity.

Authors noted “Twelve health professionals fluent in the local language (Tigrigna) were recruited.

Results

Socio-demographic characteristics of study participants

The first line statement reads “From the total of 2374 sampled healthcare providers, 2366 participants were included in the study”, I suggest to be corrected as “of 2374 participants, 2366 participated in the study” Authors are advised to revise their reports to be concise.

The first paragraph of the result section is with redundant phrases “two third is stated four times in a single paragraph”. Authors are advised to avoid redundant phrases and rephrase their result statements with eye-catching ideas to readers.

Besides authors interpreted “Nearly two-thirds of the study participants were from the Mekelle and Central zones” that mismatches with the figure on table 1 which is calculated as 26.7% for Mekelle and 15% for Central in total 41.7%. Authors are advised to make correction of the aforementioned result statement.

What standard do you for perceived economic classification?

In your result section table 2, the result of the odds ratio 0.7 [0.5, 0.9] by zone showed the protective effect of vicarious trauma among health workers in south East zone, do you have any clarification? Can you see back your data again for possible correction of the analysis?

Discussion

In the first paragraph of the discussion the word those is repeatedly stated, so authors are requested to avoid redundant words and revise the wording.

The comparison of similarities and variations in the discussion section is more skewed to the Western countries (USA, Europe and Canada). In order to make careful comparisons with sub-Saharan African countries, Authors need to revisit the discussion section.

It would be good if authors made an attempt to align some of their justifications with peer literature review.

Authors stated “Consequently, medical personnel were forced to treat a high volume of patients with the potentially dangerous physical and psychological effects of the conflict on a regular basis”. This statement is not clear. Is it part of the finding of this study, if not, Authors are requested to put a reference.

Authors stated “….. many medical experts and dangerous medical supplies needed for treatment and therapy have been forced to leave underfunded”, “What does dangerous medical supplies mean?” Authors are advised to provide clear statement on the highlighted phrase.

Many of the justifications stated on the discussion section lack references. Similar justifications are documented in various reports by humanitarian organizations including MSF, OCHA, ICRC and World Peace foundation led by Professor Alex Alex de Waal, so authors are advised to use the references for their justifications

Authors are duly requested to revisit the discussion section for further elaboration and amendment.

For the betterment of the manuscript, language revision is highly needed by native speakers.

Decision – Major revision

Reviewer #2: Dear authors,

The title is very interesting and timely to improve the healthcare service.

However, the following comments needs your attention

General comments:

• There is gloss editorial problem

• Less contextualized (context is war and war-related)

• The way research question is explored is not clear

Abstract:

• Background: Indicate clear gap

• Method does not indicate study area

• Result: show the p values for the factors

• Conclusion: disregard the conclusion about risk factors

Introduction:

• Look the detailed comments in the document

Method:

• The study setting is not clear... Conditions during the war are not indicated

• Show that your study takes place in health facilities under the study area and setting

• Specify your study population (if any inclusions and exclusions)

• Sampling procedure like stratification and proportional allocation should be clear (look the comments in detail in the document)

• Analysis method... way of reporting the OMLR... Comment in the document

Results:

• Reporting the result of ordinal logistic regression... comparison groups

• Inclusion of the zones for analysis... Concerns look comment

Discussion

• Need a huge work

• Justifications provided are not satisfactory and not supported by evidence

• Implications are not indicated

• All the narrative in the discussion should be based on scientific merit

• Strength and limitation... very shallow

Conclusion

• Has to emanate from the findings

• Conclusion: not repeating the result

Reference

• Journal articles should have year, volume, number, and page

• There are about 4 references with no year

Reviewer #3: First of all, i would like to say thank you dear plos one journal organizer for inviting me to this valuable manuscript and thanks Authors for well organized and systematically writtlng the manuscript. I moved through all parts of these manuscript and no doubt has been seen. It has acceptable comment and no more major errors are seen. Regards!

Reviewer #4: Review of War related-Vicarious Trauma among Healthcare providers in the War-Torn Tigray, Northern Ethiopia PONE-D-24-43464

This original research article is a cross sectional study of the prevalence of vicarious trauma among healthcare workers in war-torn Tigray, Northern Ethiopia. It reports highly significant levels of trauma, an important finding that could help to guide interventions to support health care workers living in this and other conflict zones with the aim of maintaining a functional health care system serving a population victimized by war. It also includes an empirically derived regression model of significant predictors of outcome that can generate hypotheses regarding the mechanism of vicarious trauma, which can help to guide intervention. I commend the authors for this important piece of research carried out despite extremely difficult circumstances. I believe it can be a very valuable addition to the peer reviewed scientific medical literature on this topic.

I believe the manuscript could be strengthened in the following specific ways prior to publication, and offer these suggestions in a supportive and constructive spirit:

1. In the title, the phrase “War related-Vicarious Trauma” should be: “War-Related Vicarious Trauma”

2. Please change “It has costed the physical, mental, economic, and psychological health of the people in Tigray in general and the healthcare providers in particular” to “It has taken a toll the physical, mental, economic, and psychological health of the people in Tigray in general and the healthcare providers in particular”

3. Please clarify the sentence: “The effect of conflicts provided a prominent example of this experience”

4. For the sentence “A systematic review revealed that 56% of health care providers had developed secondary trauma (10)” please clarify to be consistent with the other examples cited in this paragraph by briefly identifying the work setting and affected group.

5. “Maynemar” should be “Myanmar”

6. Please change “Despite our personal observations and experiences with such trauma among healthcare providers, in the war-torn Tigray region, the scope of the problem is not investigated. Except for the presence of vicarious trauma among the medical staff in the hemodialysis unit of the Ayder referral hospital published recently” to “Despite our personal observations and experiences with such trauma among healthcare providers in the war-torn Tigray region, the scope of the problem is not investigated except for a single recently published study on the presence of vicarious trauma among the medical staff in the hemodialysis unit of the Ayder referral hospital.”

7. Please define “health posts,” which is a term that may not be familiar to all readers.

8. Regarding the primary outcome measure, the “Crisis and Trauma Resource Institute's (CTRI) tool,” please specify in the text which tool was used (eg., title and if available the author(s), publisher, publication year, etc). Has the tool’s validity and reliabilty been measured and published in peer reviewed literature? If so, please cite using the publication information instead of the URL where it is found.

9. Also regarding the primary outcome measure, for the sentence stating that “After extensive revision of the English questionnaire, the final English version was translated into the local language by language experts” please briefly describe or summarize the descriptions. What needed revision?

10. In the sentence “Data collection location, accuracy, and completeness were monitored by the data manager by looking at the server-centered,” the meaning of the phrase “looking at the server-centered” is unclear. Please clarify.

11. For the sentence “The reliability of the item was measured by a Cronbach's alpha that ranged from 0.835 to 0.856” please clarify “The reliability of the item was measured by a Cronbach's alpha that ranged from 0.835 to 0.856.” Is this inter-rater reliability, something else?

12. The fact that “Variables having p ≤ 0.05 on the bivariate ordinal logistic regression analysis were candidates for the ordinal multivariable logistic regression analysis to control confounders” the fact that the ordinal logistic regression variables were derived empirically rather than from a pre-specified hypothesis means that the on average, 5% of the significance could be explained by Type I error. This should be acknowledged as a limitation of the study. This could also be addressed by use of an appropriate correction (eg., Bonferroni correction or its equivalent as statistically appropriate) in the analysis. Please either a>) re-run the data with this correction, or b.) acknowledge its absence as a further limitation. It should also be clarified that the estimated sample size and implied power analysis relate to measuring significance in the primary outcome of vicarious trauma only, not to the regression model (assuming this is the case; if not, please clarify).

13. The meaning of “than” is not clear in the sentence “When we report our results of multivariable ordinal logistic regression, we used the term ‘higher level vicarious trauma,’ which means extreme high than combined low, medium, and high; or combined ‘high and extreme high’ than combined low and medium; or combined ‘medium, high, and extreme high’ than low.” Instead of “than” do you mean “versus” (or an equivalent term such as “rather than” or “as opposed to”)?

14. By “orthodox religion” Do you mean the Christian Orthodox religion? Please clarify.

15. Regarding “The participants had a median (IQR) work experience of 8.0 (5.0, 12.0) years and a median annual income (IQR) of 8017 Ethiopian Birr (range: 7071.0–9056.0)” is this correct? I believe 7071.0–9056.0 Ethiopian Birr is equivalent to US$55.05-70.51, while the 2023 Gross Domestic Product per capita in Ethiopia was last recorded at US$890.35 US. Please confirm this is correct both in the foregoing sentence and also in Table 1.

16. Regarding “Table 2: Factors associated with vicarious trauma among healthcare providers in Tigray, Ethiopia, August to September, 2023” I suggest the authors consider making this Table 3, and adding a separate Table 2 in which data are presented on the prevalence of the different levels (low risk, moderate risk, high risk, and extreme high risk) of vicarious trauma in the sample as a whole. Alternatively, perhaps even better than a new Table 2 could be a new Figure 1 with this data represented graphically using a pie chart.

17. For the sentence “Another explanation could be that many medical experts and dangerous medical supplies needed for treatment and therapy have been forced to leave underfunded, damaged, and robbed healthcare institutions” please revise for clarity, eg: “Another explanation could be that underfunding of, damage to, and theft from healthcare institutions have led to critical shortages in medical experts and essential medical supplies needed for treatment.”

18. For the sentence, “The Tigray war, which was characterized by ethnic violence and the targeting of civilians, which medical professionals may see personally while tending to patients, this could be because the victims have experienced personal loss, have lost loved ones or families, or have been uprooted themselves” please revise for clarity, eg: “The Tigray war was characterized by ethnic violence and the targeting of civilians. Medical professionals who have experienced personal loss, have lost loved ones or families, or have been uprooted themselves may be re-traumatized when tending to patients who have also experienced such trauma.”

19. Please consider clarifying “This might mean healthcare providers working in these regions were exposed to more violence and trauma, heavy fighting during the war, shortages of medical supplies as a result of the war, may have lost colleagues, friends, or family members during the war, and may have met and provided care for people who had been injured by the crude troops of Eritrean and Amhara Milasha, like the of mass massacre in Tsion on November 29, 2020,” eg., “This might be because healthcare providers working in these regions were exposed to more violence and trauma, heavy fighting during the war, shortages of medical supplies as a result of the war, loss of colleagues, friends, or family members during the war, and/or may have met and provided care for people who had been injured by atrocities such as the of mass massacre in Tsion on November 29, 2020 by troops of Eritrean and Amhara Milasha.” Please also provide citation for this statement, eg: by human rights groups, journalists, etc.

20. Please clarify “This implication might indicate the life stage and responsibilities; this means care providers over 30 may have additional personal stressors like family obligations, more connectedness and responsibilities to nearby communities, financial concerns, or aging parents, making them less resilient to additional emotional burdens from work in addition to their traumatized clients or patients they met.” Eg., “This might be because of increasing life stage responsibilities and associated personal stressors among care providers over 30 such as family obligations, responsibilities to communities, financial obligations, or aging parents, making them less resilient to additional emotional burdens from work leading to increased risk of vicarious traumatization when caring for traumatized clients or patients.”

21. Regarding the sentence “It might also be because of their changes in coping mechanisms; this means that over time, coping strategies used earlier in a career may become less effective, and healthier coping mechanisms may not yet be developed, making individuals more vulnerable to the effects of vicarious trauma” it is not clear why coping would decrease with age.

22. Regarding the sentence “Limited personal time and reduced opportunities for self-care and emotional processing due to family commitments can exacerbate the impact of vicarious trauma” please either a.) provide a reference for this or b.) replace “can” with “might.”

23. Regarding “…but being hampered by the limited resources and seriousness of the war might lead to moral distress and feelings of helplessness, which can contribute to vicarious trauma.” Please either a.) provide a reference for this statement or b.) change “can contribute” to “might contribute.”

24. Please clarify “Additionally, it might be difficult to make decisions about resource allocation, staffing, and patient care under perilous circumstances. It might be exposed to a secret broader information about the healthcare system’s such as displaced staff, perilous medical supply, traumatized patients, and witnessing the overall impact of the war on healthcare delivery.” Eg., “Additionally, it might be difficult for managers to make decisions about resource allocation, staffing, and patient care under perilous circumstances, especially if they alone bear the burden of knowing traumatizing information about the healthcare system such as displaced staff, perilous medical supply, traumatized patients, and witnessing the overall impact of the war on healthcare delivery.”

25. Please clarify “This might be because of the nature of the war, which was totally besiege and siege.” Eg., “This might be because of the nature of the war, which involved working in areas under siege.”

26. Regarding the sentence “This would mean healthcare professionals see more critically ill patients to reach their facilities. and those whose conditions worsen due to a lack of timely treatment,” wouldn’t this traumatize all healthcare workers equally irrespective of their distance from work?

I again commend the authors for this important contribution to knowledge that can help respond to a pressing healthcare delivery service need. I thank them for their brave work, and wish them the best in their time of crisis.

Reviewer #5: This study is important in examining the vicarious trauma on the Tigray War.

Some concerns emerged.

Socio-demographic characteristics of study participants is better to be simple using Table 1.

The definition of the healthcare providers is better to be filled in the methods.

In the section of the factors Associated with Vicarious Trauma, on the aim of the analysis, what reason did the authors include the eight categories in the multivariable ordinal logistic regression mode? Are the explanations of the results is overlapping to the table 2 ?

**Do you want your identity to be public for this peer review?** For information about this choice, including consent withdrawal, please see our Privacy Policy

Reviewer #1: **Yes:** Dr. Tesfay Gebregzabher Gebrehiwet

Reviewer #2: No

Reviewer #3: **Yes:** Beshir Mammiyo

Reviewer #4: No

Reviewer #5: No

---

## [Author Response · Author response to Decision Letter 1]

17 May 2025

Responses to editor and reviewers’ comments

Dear Academic Editor-in-chief,

We are grateful to the editor and reviewers for their time and constructive comments on our manuscript. We have incorporated all the comments and suggestions and here by to submit a revised version of the manuscript using track change and point by point response letter for further consideration in the journal. As per your request here are the lists of amendments.

Journal Requirements:

Editor ’s comment

Dear Editor,We are deeply grateful for your insightful advice and positive recognition of our work. We believe the manuscript has been significantly strengthened through the incorporation of your and the reviewers' valuable feedback. Our detailed point-by-point responses to your comments are provided below.

Response to editor’s comment: Thank you so much for your guidance, we have revised our manuscript based on the PLOSE ONE journal style template and submission guideline.

Editor ’s comment

2. Your ethics statement should only appear in the Methods section of your manuscript. If your ethics statement is written in any section besides the Methods, please move it to the Methods section and delete it from any other section.

Response to editor’s comment: Thank you so much for your suggestion, the ethics statement is included in the methods section following data processing and analysis.

Editor ’s comment

3. We note that you have indicated that there are restrictions to data sharing for this study. For studies involving human research participant data or other sensitive data, we encourage authors to share de-identified or anonymized data. However, when data cannot be publicly shared for ethical reasons, we allow authors to make their data sets available upon request. For information on unacceptable data access restrictions, please see

Response to editor’s comment: Thank you so much for your comment, as far as our data set is concerned, there is no any restrictions, and we are able to submit the minimally anonymized data set as an additional file.

Editor ’s comment

4. Your abstract cannot contain citations. Please only include citations in the body text of the manuscript, and ensure that they remain in ascending numerical order on first mention.

Response to editor’s comment: Thank you so much for your suggestion, but our abstract does not include citations. We apologize if the numbers we included in the method section, which shows the vicarious trauma score as low risk (0–14), moderate risk (15–21), high risk (22-28), and extremely high risk (29–35), are making confusion.

Editor’s comment

5. Figure 1 in your submission contain [map/satellite] images, which may be copyrighted. All PLOS content is published under the Creative Commons Attribution License (CC BY 4.0). We require you to either (1) present written permission from the copyright holder to publish these figures specifically under the CC BY 4.0 license, or (2) remove the figures from your submission.

Response to editor’s comment: Thank you so much for your comment, we discussed the choices you mentioned in comment number five, and we decided to delete figure one from the manuscript.

Editor’s comment

6. We note you have included a table to which you do not refer in the text of your manuscript. Please ensure that you refer to Table 2 in your text; if accepted, production will need this reference to link the reader to the Table.

Response to editor’s comment: Thank you so much for your comment, Table 2 has been included on the manuscript at the end of the text description, where to view at the end sentence of page 9.

Additional Editor specific Comments:

Abstract

Editor’s comment: Avoid third-person writing “the”

Response to editor’s comment: Thank you so much for your suggestion, we have revised the article “the” and made some modifications

Editor’s comment: VT seems difficult to understand so it is better to define it in the abstract as well as in one sentence.

Response to editor’s comment: Thank you so much for your suggestion, we have revised the abstract, we wrote in full sentence as vicarious trauma and add one sentence “Vicarious trauma refers to harmful changes that occur in professionals’ views of themselves and/or others as a result of deep empathic engagement and repeated exposure to details of trauma survivor”.

Editor’s comment: “we estimated “third person writing the whole sentence needs rephrasing

Response to editor’s comment: Thank you so much for your comment, we tried to revise the whole sentence and make an edition.

Editor’s comment: The burden of war and its impact especially on those with health providers, what is the need to do especially health care workers

Response to editor’s comment: Thank you so much for your comment, in conflict zones, humanitarian agencies emphasize the critical need to safeguard healthcare workers so they can continue their life-saving work, International Humanitarian Law (IHL) and respect for healthcare; agencies like the ICRC, WHO, and Safeguarding Health in Conflict Coalition (SHCC) stress that healthcare workers and facilities are protected under the Geneva Conventions and additional protocols. Attacks against them are war crimes.

Editor’s comment: I would put AOR of the factors significantly associated

Response to editor’s comment: Thank you so much for your suggestion; we put the value of AOR for the factors significantly associated.

Introduction

Editor’s comment: Line 3-5 can you quantify the material or human damage with reference

Response to editor’s comment: Thank you so much for your suggestion; the total economic cost of the war-related looting or vandalism in monetary terms was more than $3.78 billion, and the damage to the economic value in monetary terms was more than $2.31 billion. Meanwhile, the direct economic loss to the health system in monetary terms was more than $511 million. According to this assessment, 514 (80.6%) health posts, 153 (73.6%) health centers, 16 (80%) primary hospitals, 10 (83.3%) general hospitals, and 2 (100%) specialized hospitals were damaged and/or vandalized either fully or partially due to the war. Reference kindly look at the articles “damage to the public health system caused by war-related looting or vandalism in the Tigray region of Northern Ethiopia” cited as (Gufue ZH et al., Front. Public Health 12:1271028. doi: 10.3389/fpubh.2024.1271028).

Editor’s comment: Citation for the “deliberate targeting of war on health care system causing disruption

Response to editor’s comment: Thank you so much for your comment; we put the reference for this sentence which was paraphrased from the sentence“Many have reported that the deliberate destruction, vandalisation and looting of the entire health system have been the hallmarks of the ongoing conflict” published by Hailay Gesesew and his colleagues cited as (Gesesew H, Berhane K, Siraj ES, et al., BMJ Global Health 2021;6:e007328. doi:10.1136/ bmjgh-2021-007328).

Editor’s comment:

• Paragraph 3; why healthcare professionals; are at risk needs rephrasing and synthesis,

• Rather than listing all studies to connect ideas and flow in paragraphs 3 and 4

• First discussing issues in Western then Africa rather than shifting back and forth

Response to editor’s comment: Thank you so much for your comment; we rewrite paragraph 3 and 4, where to look paragraph 3 on the “revised manuscript with track and manuscript” attached.

Methodology

Editor’s comment: You need to describe the desire why you used p 50 % inside the text and design effect 2; need justification.

Response to editor’s comment: Thank you so much for your comment; we used p 50% because there were no related studies in Ethiopia and taking prevalence from other countries might not represent the situation in Tigray. Thus, we had used p 50%. Regarding the design effect of 2, as a rule of thumb design effect of 2 or 1.5 is frequently used in sampling, especially for cluster, multistage, and stratified designs because of their stages and intracluster correlation, to provide a practical way to inflate the sample size and account for the potential loss of precision compared to simple random sampling. However, it's essential to recognize its limitations and strive for more precise estimates of the design effect whenever possible.

Editor’s comment: If you put a diagram and the specific numbers when doing multi-stage sampling and proportional allocation

Response to editor’s comment: Thank you so much for your comment; we put the diagrammatic presentation of sampling technique and uploaded as Figure 1.

Editor’s comment: What other tools or factors are assed other than VT, which is not put in the methodology

Response to editor’s comment: Thank you so much for your suggestion; we add on the methodology all the rest factors such as sociodemographic, and work area related characteristics. Editor’s comment: Why do you use a p-value 0.05 for bivariate logistic regression and put the justification as well

Response to editor’s comment: Thank you so much for your suggestion; recognizing the absence of a definitive threshold for transitioning variables from bivariate to multivariate analysis, our study, leveraging a substantial sample size that inherently yielded a high variable-to-case ratio, strategically advanced all variables exhibiting statistical significance (p < 0.05) in the bivariate logistic regression to the subsequent multivariate model. This approach was deliberately employed to rigorously control for potential confounding influences and to examine the relationship between independent and dependent variables.

Result

Editor’s comment: You can also put the reliability of the tools of outcome and validation in Ethiopia.

Response to editor’s comment: Thank you so much for your comment; to ensure the robustness of our data collection, the reliability of the tool was rigorously assessed within the Ethiopian setting. This evaluation, conducted as part of our methodological approach, demonstrated strong internal consistency, evidenced by a Cronbach's alpha of 0.856 and we had putted at the “data quality control section at the end of the paragraph.

Editor’s comment: The footnotes a, b,c are confusing as you only used p-value 0.05 as statically significant in your methodology.

Response to editor’s comment: Thank you so much for your comment; the statistical significance of the identified factors is denoted by superscript letters, where 'a' indicates a p-value ranging from 0.05 to 0.01, 'b' signifies a p-value between 0.01 and 0.001, and 'c' represents a highly significant p-value of less than 0.001. If seems not important to show the level of significance we can remove.

Editor’s comment: Why you did not asses the type of traumatic content disclosed to health providers

Response to editor’s comment: Thank you so much for your comment; during the development of our proposal, we made a conscious decision to avoid directly questioning healthcare providers about the specific traumas they had experienced. This decision was based on several factors: (1) the potential for multiple trauma exposure would make it difficult to pinpoint the source of vicarious trauma; (2) the varied trauma histories of patients could obscure the link to specific instances of vicarious trauma; and (3) eliciting detailed accounts of past traumas could inadvertently exacerbate vicarious trauma symptoms through recall and reflection during the data collection process.

Editor’s comment: Consequently, medical personnel were forced to treat a high volume of patients with the potentially dangerous physical and psychological effects of the conflict on a regular basis. Another explanation could be that many medical experts and dangerous medical supplies needed for treatment and therapy have been forced to leave underfunded, damaged, and robbed healthcare institutions. This places a great deal of strain on the remaining employees, who are already probably overburdened and worn out, making them even more susceptible to vicarious stress. The Tigray war, which was characterized by ethnic violence and the targeting of civilians, which medical professionals may see personally while tending to patients, this could be because the victims have experienced personal loss, have lost loved ones or families, or have been uprooted themselves. This may exacerbate their inability to detach themselves from their traumatized patients and heighten their sense of helplessness and hopelessness.” This concept needs to be back up by citation and references

Response to editor’s comment: Thank you so much for your comment; based on the authors' direct experiences during the war, and supported by the literature cited in references (19, 20), this paragraph outlines the difficult circumstances faced for healthcare providers.

Editor’s comment: The discussion needs to be improved as it does not compare or give appropriate citations for the explanation of the discussion points.

Response to editor’s comment: Thank you so much for your comment; we revised the discussion and corrected on the manuscript.

Editor’s comment: I would add a specific and better recommendation.

Response to editor’s comment: Thank you so much for your comment; we also tried to revise the recommendation.

Response to Reviewer #1

Reviewer suggestion: I hereby submit a review report of the manuscript, entitled ‘War related -Vicarious Trauma among Healthcare providers in the War-Torn Tigray, Northern Ethiopia”. I acknowledge the efforts of the authors for performing a good job and writing evidence, which is highly needed. Since, it needs further amendment, I suggested to be accepted with Major revision. I believe, this manuscript will be very helpful in passing better evidence for decision making in rebuilding the human resource development.

Response to reviewer suggestion: Dear Reviewer, we are deeply grateful for your insightful advice and positive acknowledgment of our work. We believe the manuscript has been significantly strengthened through the incorporation of all the valuable feedback from you and the editor. Our point-by-point responses to your specific comments are detailed below.

Reviewer comment: For the sake of simplicity, it would be good to create line numbers of the manuscript, which would be easier for the authors to trace the comments and revised changes.

Response to reviewer comment: Thank you so much for your suggestion; we put line number on the manuscript.

Abstract

Reviewer comment: On the method section of the abstract a statement is described as “VT was assessed using a 7-item standard tool with likert scale…”, a phrase in a new line statement shall be written with full words of the phrase, otherwise, authors are advised to write Vicarious Trauma instead of VT

Response to reviewer comment: We appreciate your comment and have used full words throughout the abstract, particularly on new lines.

Reviewer comment: On the second line of the result section of the abstract, authors need to rephrase the “was found to be”.

Response to reviewer comment: We are grateful for your comment and have revised the phrase based on your suggestion.

Reviewer comment: The conclusion of the abstract seems overlooked.

Response to reviewer comment: Thank you so much for your comment; based on your input, we undertook a revision of the conclusion.

Introduction

Reviewer comment: On the first and second paragraphs: Authors have verified about the war in Tigray and its effects on the vulnerability of the health workers referring three peer reviewed articles. Ample literature written by humanitarian organizations have proven the health workforce traumatic experiences, so authors are requested to justify their statements from the perspective of humanitarian agencies.

Response to reviewer comment: Thank you so much for your comment; we have added some evidence from the humanitarian agencies, where to find them the whole paragraph 2.

Reviewer comment: Third and fourth paragraph, the piece of evidences from Gaza, Poland, Eastern Maynemar, Kenya, USA, Japan and findings from University of Ottawa needs to be consolid

---

## [Decision Letter · Decision Letter 1]

1 Jul 2025

Dear Dr./Mr.  Hagos Degefa Hidru,

plosone@plos.org

We look forward to receiving your revised manuscript.

Kind regards,

Selamawit Alemayehu Tessema

Guest Editor

PLOS ONE

Journal Requirements:

Additional Editor Comments (if provided):

Thank you for revising your paper with comments and suggestions. However, these areas need further editing.

Abstract

•The burden of war and its impact, especially on those with health providers, what is the need to do, especially for healthcare workers

•Interventions such as aftercare and addressing the vulnerable groups

•It would be better to add relevant recommendations on the conclusions based on the findings

Introduction

•It is good that you put references, but on lines 3-5, can you quantify in numbers the material or human damage?

•Paragraph 3; why health care professionals are at risk needs rephrasing and synthesis

•Rather than listing all studies to connect ideas and flow in paragraphs 3 and 4

•First discussing issues in western then Africa, rather than shifting back and forth.

•More details on the war in Ethiopia and its impacts, before going to the scarcity of data

Methodology

•You need to describe the reason why you used p 50 % inside the text, and the design effect 2; need justification; no justification in the revised version.

•Why did you use p p-value of 0.05 for bivariate logistic regression and put the justification as well.

•Sensitive topic, more clarification on ethical considerations

Result

•The footnotes a, b,c are confusing, as you only used p p-value of 0.05 as statistically significant in your methodology.

• Not including the type of traumatic limitations as well disclosed to health providers.

•The discussion needs to be improved as it does not compare or give appropriate citations for the explanation of the discussion points.

•It is good to add specific recommendations based on the findings.

•Thorough edition of grammar and spelling

Reviewers' comments:

Reviewer's Responses to Questions

**Comments to the Author**

Reviewer #1: All comments have been addressed

Reviewer #4: All comments have been addressed

Reviewer #5: All comments have been addressed

2. Is the manuscript technically sound, and do the data support the conclusions?

Reviewer #1: Yes

Reviewer #4: Yes

Reviewer #5: Yes

3. Has the statistical analysis been performed appropriately and rigorously?

Reviewer #1: Yes

Reviewer #4: Yes

Reviewer #5: Yes

4. Have the authors made all data underlying the findings in their manuscript fully available?

Reviewer #1: Yes

Reviewer #4: Yes

Reviewer #5: Yes

5. Is the manuscript presented in an intelligible fashion and written in standard English?

Reviewer #1: Yes

Reviewer #4: Yes

Reviewer #5: Yes

Reviewer #1: Dear Academic Editor and authors

I am very glad to re-review the manuscript “War-related Vicarious Trauma among Healthcare providers in the War-Torn Tigray, Northern Ethiopia”

I re-reviewed the revised manuscript and it is well written except few typographic and editorial errors. The revised version of the manuscript showed much improvement. Since the evidence is highly needed for decision making and restoration of the human resource development, I suggest this revised manuscript to be accepted after the minor errors are corrected.

Minor comments that need to be considered are below:

Line 41: Authors need to edit word spacing.

Line 51-52: Method of the abstract section: Authors wrote: “Statistical significance was reported whenever the 95% confidence interval for odds ratio did not include one and the p-value was less than 0.05”. This statement contradicts with their finding on line 235-236 Table 2: Factors associated with vicarious trauma among healthcare providers in Tigray; for instance in table 2, the variable age in years “age group 20-29 is reference, then the vicarious trauma for the age groups 30-39, 40-49 and >50 showed results of AOR and CI 1.3 [1.1, 1.6], 1.5 [1.1, 2.0] and 2.3 [1.7, 3.0]. In this case the confidence intervals include 1 which contradicts to the statement reported by authors. Therefore, Authors need to reconsider their statement related to statistical significance based on the valid and scientific description.

Line 95 and 99: Authors need to edit word spacing.

Authors wrote sometimes VC in abbreviation and others in full words; please check Line 98, 100, 101, 104, 105, 110. Authors are advised to be consistent in writing the phrase.

Line 114: Authors need to state “health post” in plural form as “health posts”.

Line 199-200: Authors stated: 66.1%, of the 200 respondents were living with their parents. It is uncommon to start with numbers in the new line statement, so authors are advised to rewrite the statement: of the 200 respondents, two third/or 66.1% were living with their parents.

Line 328: Authors wrote “………………consequence of the Tigary war among healthcare professional”. The bolded word should be written in plural form. “professionals”

Reviewer #4: (No Response)

Reviewer #5: (No Response)

**Do you want your identity to be public for this peer review?** For information about this choice, including consent withdrawal, please see our Privacy Policy

Reviewer #1: **Yes:** Dr. Tesfay Gebregzabher Gebrehiwet

Reviewer #4: No

Reviewer #5: No

---

## [Author Response · Author response to Decision Letter 2]

22 Jul 2025

Responses to editor and reviewers’

We are writing to express our sincere gratitude to both the editor and the reviewers for your invaluable time, insightful comments, and the positive reception of our previous responses. Your feedback has been instrumental in significantly enhancing the quality of our manuscript and reached at this stage. We have carefully incorporated all the additional comments and suggestions you provided. Accordingly, we are pleased to resubmit a revised version of our manuscript and manuscript with tracked changes for your easy reference, along with a detailed point-by-point response letter addressing each comment. As requested, a comprehensive list of all amendments is included for your further consideration. We genuinely appreciate your continued support. Thank you so much.

Journal Requirements:

Dear Editor, We are deeply grateful for your valuable advice and positive assessment of our manuscript. We feel the revisions, based on your and the reviewers' feedback, have significantly improved the work. Below, you will find our point-by-point responses to your few comments. We're very grateful for your detailed and insightful advice.

Editor’s comment

Please review your reference list to ensure that it is complete and correct. If you have cited papers that have been retracted, please include the rationale for doing so in the manuscript text, or remove these references and replace them with relevant current references.

Response to editor’s comment: Thank you so much for giving helpful suggestion. We have incorporated those references cited within the text.

Additional Editor Comments (if provided):

Editor’s comment

Thank you for revising your paper with comments and suggestions. However, these areas need further editing.

Response to editor’s suggestion: We are immensely grateful for your thoughtful feedback and the significant encouragement you have given us on this paper. It means a great deal to us. We are enthusiastic, thank you in advance.

Abstract

Editor’s comment

The burden of war and its impact, especially on those with health providers, what is the need to do, especially for healthcare workers, interventions such as aftercare and addressing the vulnerable groups, it would be better to add relevant recommendations on the conclusions based on the findings

Response to editor’s comment: We are deeply appreciated for your insightful suggestion; we add some specific recommendation based on our findings. Thank you a lot.

Introduction

Editor’s comment

It is good that you put references, but on lines 3-5, can you quantify in numbers the material or human damage?

Response to editor’s comment: Thank you so much for your comment, we have add the sentence related to the material damage on the manuscript “The total economic cost of the war-related looting or vandalism in monetary terms was more than $3.78 billion, and the damage to the economic value in monetary terms was more than $2.31 billion’ (Gufue ZH, et al., April 5, 2024). Editor’s comment

Paragraph 3; why health care professionals are at risk needs rephrasing and synthesis and Rather than listing all studies to connect ideas and flow in paragraphs 3 and 4.

Response to editor’s comment: Thank you for your insightful comment and it's greatly appreciated. We have rephrased and rewritten paragraph 3, and critically interconnected the ideas and strengthened the flow between paragraphs 3 and 4. These changes are visible in both the manuscript and the manuscript with track-change we have attached.

Editor’s comment: First discussing issues in western then Africa, rather than shifting back and forth.

Response to editor’s comment: Thank you so much for your suggestion, we have revised and tried to put from global to local (Western to Africa).

Editor’s comment: More details on the war in Ethiopia and its impacts, before going to the scarcity of data

Response to editor’s comment: We are highly appreciated for your nice insightful, thank you so much. We have revised and add paragraph 5 before directly going to the scarcity of the data.

Methodology

Editor’s comment: You need to describe the reason why you used p 50 % inside the text, and the design effect 2; need justification; no justification in the revised version and why did you use p-value of 0.05 for bivariate logistic regression and put the justification as well.

Response to editor’s comment: We deeply appreciate your feedback and have now incorporated our responses into the manuscript. Thank you so much.

Result

Editor’s comment: The footnotes a, b, c, are confusing, as you only used p-value of 0.05 as statistically significant in your methodology

Response to editor’s comment: We appreciate your kind suggestion. To be consistent with the methodology section we have specified the level of statistical significance, we have removed the footnotes. Previously, we had only included the variables to show their level of significance.

Editor’s comment: Not including the type of traumatic limitations as well disclosed to health providers.

Response to editor’s comment: We truly appreciate your very insightful comment. As we have tried to mention within our study's limitations, information regarding the specific types and durations of trauma exposure were not included in this study.

Editor’s comment: The discussion needs to be improved as it does not compare or give appropriate citations for the explanation of the discussion points

Response to editor’s comment: We highly appreciated for your comment; we have tried to revise the discussion part.

Editor’s comment: It is good to add specific recommendations based on the findings

Response to editor’s comment: We greatly appreciated for your insightful comment; based on your suggestion we have revised the recommendation and add some recommendations addressed specific activities.

Editor’s comment: Thorough edition of grammar and spelling

Response to editor’s comment: Thank you so much for your comment; we have tried to edit the grammar and spelling errors support by online free grammar checkers like quillbot grammar checker.

Response to Reviewer #1

Reviewer suggestion: Dear Academic Editor and authors, I am very glad to re-review the manuscript “War-related Vicarious Trauma among Healthcare providers in the War-Torn Tigray, Northern Ethiopia” I re-reviewed the revised manuscript and it is well written except few typographic and editorial errors. The revised version of the manuscript showed much improvement. Since the evidence is highly needed for decision-making and restoration of the human resource development, I suggest this revised manuscript to be accepted after the minor errors are corrected.

Response to reviewer suggestion: Dear Reviewer, we greatly appreciated for your insightful advice and positive acknowledgment of our work. Thus, your comments were found much more importance to reach our manuscript this stage of improvement among with the Editors and other reviewers’ comment, thank you so much. Here below we put our point-by-point responses to your specific comments;

Minor comments that need to be considered are below:

Reviewer comment: Line 41: Authors need to edit word spacing.

Response to reviewer comment: Thank you so much for your suggestion; we have edited space thank you so much for nice input.

Reviewer comment: Line 51-52: Method of the abstract section: Authors wrote: “Statistical significance was reported whenever the 95% confidence interval for odds ratio did not include one and the p-value was less than 0.05”. This statement contradicts with their finding on line 235-236 Table 2: Factors associated with vicarious trauma among healthcare providers in Tigray; for instance in table 2, the variable age in years “age group 20-29 is reference, then the vicarious trauma for the age groups 30-39, 40-49 and >50 showed results of AOR and CI 1.3 [1.1, 1.6], 1.5 [1.1, 2.0] and 2.3 [1.7, 3.0]. In this case the confidence intervals include 1 which contradicts to the statement reported by authors. Therefore, Authors need to reconsider their statement related to statistical significance based on the valid and scientific description.

Response to reviewer comment: We really appreciate your insightful; we attempted to explain that the AOR 95% CI does not contain the null value, which is 1.0. Any value above this, including its decimal value of 1.05, 1.1, 1.2, etc., is considered significant. However, in order to prevent ambiguity, we have utilized the p-value to indicate the level of significance instead of using such a statement.

Reviewer comment: Line 95 and 99: Authors need to edit word spacing

Response to reviewer comment: We truly appreciate your suggestion. We have corrected this word space.

Reviewer comment: Authors wrote sometimes VT in abbreviation and others in full words; please check Line 98, 100, 101, 104, 105, 110. Authors are advised to be consistent in writing the phrase.

Line 114: Authors need to state “health post” in plural form as “health posts”.

Response to reviewer comment: We are happy by your insightful for your comment; based on your suggestion we have revised and edited the whole document to be consistent.

Reviewer comment: Line 199-200: Authors stated: 66.1%, of the 200 respondents were living with their parents. It is uncommon to start with numbers in the new line statement, so authors are advised to rewrite the statement: of the 200 respondents, two third/or 66.1% were living with their parents.

Response to reviewer comment: Thank you so much for your nice comment; we have revised the sentence as your direction.

Reviewer comment: Line 328: Authors wrote “………………consequence of the Tigary war among healthcare professional”. The bolded word should be written in plural form. “professionals”

Response to reviewer comment: Thank you so much for your suggestion, we have revised as you suggested.

Response to Reviewer #4 and #5

Dear reviewers, we greatly appreciated for your insightful comments and suggestion you have had given on the previous comment that help us to develop this improved manuscript. Thank you so much all.

---

## [Decision Letter · Decision Letter 2]

30 Sep 2025

Dear Dr. Hidru,

Thank you for submitting your manuscript to PLOS ONE. After careful consideration, we feel that it has merit but does not fully meet PLOS ONE’s publication criteria as it currently stands. Therefore, we invite you to submit a revised version of the manuscript that addresses the points raised during the review process.

We look forward to receiving your revised manuscript.

Kind regards,

Selamawit Alemayehu Tessema

Guest Editor

PLOS ONE

Journal Requirements:

Reviewers' comments:

Reviewer's Responses to Questions

**Comments to the Author**

Reviewer #6: (No Response)

2. Is the manuscript technically sound, and do the data support the conclusions?

Reviewer #6: Yes

3. Has the statistical analysis been performed appropriately and rigorously?

Reviewer #6: Yes

4. Have the authors made all data underlying the findings in their manuscript fully available?

Reviewer #6: Yes

5. Is the manuscript presented in an intelligible fashion and written in standard English?

Reviewer #6: No

Reviewer #6: Thank you for writing this important article of vicarious trauma among Healthcare providers in the War-Torn Tigray, Northern Ethiopia. My comments are provided below:

1. Check grammar on line 72-75.

2. Typo word "involvement" in line 89

3. Typo word "similarly" in line 103

4. Line 157 - use it is instead of "it's" in formal writing

5. Methodology: there is no report of the validity value of the tool used

6. Typo word "prevalence" in line 236

7. Sometimes the word "moderate" is used (line 237), sometimes the word "medium" is used (Table 2): make sure the term used is consistent.

8. Typo "agencies" line 279

9. Improve grammar and comprehension in line 288-293.

10. Line 324-350 must have references to justify the claims

11. There is no consistency in the format of the references, font sizes across the texts

**Do you want your identity to be public for this peer review?** For information about this choice, including consent withdrawal, please see our Privacy Policy

Reviewer #6: No

---

## [Author Response · Author response to Decision Letter 3]

24 Oct 2025

Responses to editor and reviewers’

We are writing to express our deep gratitude to both the editor and the reviewers for your invaluable time, very insightful comments, and the positive reception of our previous responses. Dear all your feedback has been significantly building and enhancing the quality of our manuscript and reached at this stage. We have carefully incorporated all the additional comments and suggestions you provided. Accordingly, we are pleased to resubmit a revised version of our manuscript and manuscript with tracked changes for your easy reference, along with a detailed point-by-point rebuttal letter addressing each comment. As requested, a comprehensive list of amendments is included for your further consideration. We genuinely appreciate your continued support for this manuscript to reach at this stage. Thank you so much for your invaluable time and strong scientific comments, suggestions, and guidance.

Response to Reviewer #6

Reviewer suggestion: Thank you for writing this important article of vicarious trauma among Healthcare providers in the War-Torn Tigray, Northern Ethiopia. My comments are provided below.

Response to reviewer suggestion: Dear reviewer, we greatly appreciated for your kind words and positive acknowledgment of our work. Thus, your comments were found much more importance to reach our manuscript this stage of improvement among with the Editors and other reviewers’ comment, thank you so much for your insightful comments and suggestions you provided. Here below we put our point-by-point responses to your specific comments:

Reviewer comment: Check grammar on line 72-75.

Response to reviewer comment: Thank you so much and we greatly appreciated for your nice comment, we have revised the grammar of the above paragraph and edited.

Reviewer comment: Typo word "involvement" in line 89

Response to reviewer comment: We really appreciate your insightful comment; we have corrected the typo error and wrote as “involvement”.

Reviewer comment: Typo word "similarly" in line 103

Response to reviewer comment: We truly appreciate your nice comment. We have corrected this typo error as “similarly”

Reviewer comment: Line 157 - use it is instead of "it's" in formal writing

Response to reviewer comment: We greatly appreciated for your insightful comment; based on your comment we have edited and wrote as “it is”.

Reviewer comment: Methodology: there is no report of the validity value of the tool used Response to reviewer comment: Thank you in advance for your insightful comment. We have revised and added the validity report of our measurement tool. Where to look on the data quality control line 186-188

Reviewer comment: Typo word "prevalence" in line 236

Response to reviewer comment: Thank you so much for your, we are really very delighted for your comments and we have corrected and wrote as “prevalence”.

Reviewer comment: Sometimes the word "moderate" is used (line 237), sometimes the word "medium" is used (Table 2): make sure the term used is consistent.

Response to reviewer comment: Thank you so much for your insightful comment, we have revised and corrected as moderate instead of the word medium.

Reviewer comment: Improve grammar and comprehension in line 288-293

Response to reviewer comment: Thank you so much and greatly appreciated for your comment, we have rewrote and paraphrasing the paragraph. Where to look at the manuscript with track changes line 292-304.

Reviewer comment: Line 324-350 must have references to justify the claims

Response to reviewer comment: Thank you so much for your insightful comment. For this justification, we did not put a reference because this justification is the researchers’ point of view; just using their biological plausibility, they had stated the possible reason why families having larger family size had experienced more vicarious trauma.

Reviewer comment: There is no consistency in the format of the references, font sizes across the texts

Response to reviewer comment: Thank you so much for your great and insightful comment, we have revised and corrected the whole references its font size and other error.

---

## [Editor Report · Decision Letter 3]

30 Dec 2025

War-related Vicarious Trauma among Healthcare providers in the War-Torn Tigray, Northern Ethiopia

PONE-D-24-43464R3

Dear Dr. Hagos Degefa,

We’re pleased to inform you that your manuscript has been judged scientifically suitable for publication and will be formally accepted for publication once it meets all outstanding technical requirements.

Kind regards,

Selamawit Alemayehu Tessema

Guest Editor

PLOS One

Additional Editor Comments (optional):

Authors have addressed all the suggestions made by reviewer.

The manuscript may benefit from  proof reading and make edits only.

Authors need to work on this issues before publication;

-editing this sentence in the abstract to be meaningful "Due to the lack of evidence regarding the level of vicarious trauma in this population"

-need to put citation on the lines 106-110

- need to support the last two paragraphs of the discussion with other literature in lines 337-363
---

## [Editor Report · Acceptance letter]

PONE-D-24-43464R3

PLOS One

Dear Dr. Hidru,

I'm pleased to inform you that your manuscript has been deemed suitable for publication in PLOS One. Congratulations! Your manuscript is now being handed over to our production team.

Kind regards,

on behalf of

Dr. Selamawit Alemayehu Tessema

Guest Editor

PLOS One